# Review on Distribution Network Optimization under Uncertainty

Huilian Liao[ID]

Power, Electrical and Control Engineering, Sheffield Hallam University, Sheffield S1 1WB, UK;
huilian.liao@shu.ac.uk

**Abstract:** With the increase of renewable energy in electricity generation and increased engagement from demand sides, distribution network planning and operation face great challenges in the provision of stable, secure and dedicated service under a high level of uncertainty in network behaviors. Distribution network planning and operation, at the same time, also benefit from the changes of current and future distribution networks in terms of the availability of increased resources, diversity, smartness, controllability and flexibility of the distribution networks. This paper reviews the critical optimization problems faced by distribution planning and operation, including how to cope with these changes, how to integrate an optimization process in a problem-solving framework to efficiently search for optimal strategy and how to optimize sources and flexibilities properly in order to achieve cost-effective operation and provide quality of services as required, among other factors. This paper also discusses the approaches to reduce the heavy computation load when solving large-scale network optimization problems, for instance by integrating the prior knowledge of network configuration in optimization search space. A number of optimization techniques have been reviewed and discussed in the paper. This paper also discusses the changes, challenges and opportunities in future distribution networks, analyzes the possible problems that will be faced by future network planning and operations and discusses the potential strategies to solve these optimization problems.

**Keywords:** distribution network optimization; flexibility exchange; demand side management; network planning and operation

## 1. Introduction

Distribution networks face great challenges with the changes in current and future distribution networks, such as the inclusion of more green energy, installation of more controllable power electronic devices, differentiated power quality requirements from different customers and increased active engagement from customer sides. To provide stable and greener electricity and meet the requirements from various stakeholders, the network should properly plan and utilize the available network resources to meet the constraints, improve quality of services and reduce the operating cost. Proper planning/operation strategies enable the cost-effective running of the network and improved customer experience in using electricity or participating in network operation/management. Distribution planning and operation problems (such as the integration of more renewable energy, the utilization of flexibility resources and customer engagement for various purposes, etc.) can be tackled with appropriate definition of optimization problems and the use of properly tailored optimization techniques. The research on distribution system optimization can be broadly divided into two categories:

(1) Planning: With the global trend of using more renewable energy to reduce emission, one of the challenges in distribution system planning nowadays is to integrate more distributed energy

resources in existing networks by finding the optimal sizes of distributed generators (DG) and their installation locations (Section 3.2) while ensuring stable network operation [1]. With the increased load demand, aging facilities and limited network capacity, power quality (PQ) phenomena and constraint violation cause great financial loss to both Transmission System Operators (TSOs) and customers, thus proper and optimal installation of PQ mitigation devices is needed in order to provide sufficient power quality to customers (Section 3.3). Distribution system planning also looks into optimal meter placement [2] for the improvement of accuracy in state estimation (Section 3.1), and optimal strategy of network expansion/reinforcement in order to increase network capacity and facilitate network changes [3], among other factors.

(2)  Operation: This involves the daily management and operation in utilization of network analysis and optimization [4]. Operation becomes more challenging than ever because of the high penetration of renewable resources in networks, e.g., photovoltaic (PV) generation and wind turbines) [5]. The renewable energy sources in nature are highly stochastic and intermittent depending on the weather conditions. Without proper operation strategies, these renewable resources can cause instability and power quality issues in networks, such as unbalance phenomena and violation of thermal limits of the grid with high ramping in voltages and currents. Proper constraint management is required to ensure the network states within an acceptable range (Section 4.1). With the new increased flexibility and controllability, the resources in the network (including DG and load flexibility) can be utilized to achieve certain purposes, such as constraint management and solving congestion issues (Section 4.2).

With the changes and new features of current and future distribution networks, there is a great uncertainty in network conditions, including the uncertain DG outputs caused by the intermittency of renewable energy, uncertain customer behaviors in electricity use—especially with the increasingly active impact of various stakeholders on network operation and uncertain credibility of various data sources (Section 2). These uncertainties result in great fluctuation and unpredictability and impose great challenges to network planning and operation. These uncertainties have to be addressed in optimization in order to generate better planning and operation strategies that are suitable to realistic power networks (Section 3.4).

This paper focuses on a number of critical optimization problems (distribution planning and operation) in the network level, including introduction to the definition of the optimization problems and the design of optimization frameworks in order to accurately and efficiently search for the optimal strategies. This paper identifies the critical uncertainties in distribution optimization and investigates the approaches for addressing differentiated uncertainties in network analysis and performance assessment. Section 2 reviews the data resources that can be used for distribution network planning and operation, and also identifies the uncertainties of the data sources that should be considered in optimization. Section 3 discusses on the optimization-based planning in distribution networks, including optimal meter placement for improving network analysis functions, optimal DG planning for maximizing DG integration and optimal placement of mitigation devices for power quality mitigation. Section 3 specifically discusses the approaches of making use of the prior-knowledge of power system infrastructure in order to improve the efficiency and accuracy of optimization searching. It provides guidance on the design of an optimization framework under the context of power systems. Section 4 investigates on the optimization problems in distribution operation, including constraint management and optimal strategies for flexibility exchange. For Sections 3 and 4, a wide range of optimization techniques used in distribution optimization has been introduced and compared. In Section 5 this paper also presents the possible future network optimization problems and discusses the potential solutions to solve these problems, such as the event-triggered approach to solve the flood of a huge data stream, optimal provision of differentiated power quality to meet different customers' requirements and data self-correction using the "close-loop" information flow framework.

## 2. Measurement and Uncertainty

Data are the most important pre-conditions for completing a number of critical functions in power systems planning, management and operation. The accuracy and representativeness of the data determines the quality of the final solutions obtained in the optimization process. In distribution networks, data can be achieved from various sources which have different levels of accuracies and uncertainties. The different uncertainties usually should be addressed in the optimization process via assigning differentiated confidence/weights to the corresponding data. The use of a proper way of dealing with uncertainties can overall improve the performance of the functions/algorithms to some extent. Generally data can be mainly collected either directly or indirectly.

### 2.1. Measurements

Data can be obtained directly from meters installed across the networks. These data measurements provide the most accurate information that can be used in different network functions. However, since there are many busbars and lines in distribution systems, it is impossible to install sufficient meters to achieve full network observability. Therefore, other data sources should be sought to minimize the issues caused by insufficient real meter measurements. The impact of lacking real measurements on optimization/application performance can be minimized by the incorporation of pseudo-measurements (PMs), including mixed model pseudo-measurements, scheduled power and load estimation. To obtain better PMs, various data sources in distribution networks have been explored for network analysis [6]. Nowadays in particular, the development of smart concepts in distribution networks enables the collection and storage of massive amounts of historical data (or on-line data) in different forms. Indirect measurements (also called pseudo-measurement) can be gathered via data analytic approaches, such as estimation and forecasting. Although these data sources provide lower accuracy and higher uncertainty of information compared to real meter measurements, the network analysis or estimation performance can be significantly improved by properly addressing the differential influence of various indirect data sources in decision-making. For instance, the detailed historical data of load demands is available thanks to widely distributed smart meters. PMs of load demand can be extracted and estimated based on non-synchronized measurements obtained from smart meters using load estimation techniques. A number of studies have investigated the use of smart meter data to improve the estimation of various parameters in low voltage distribution networks [6–9]. Computation intelligence methods have been used to generate PMs of load demands. For instance, machine learning approaches were used to yield reliable inputs for State Estimation (SE) [10] and artificial neural networks used to generate load demands [11]. Alternatively, the real-time load estimation can be carried out by the interaction between estimation and load flow [12]. In [13], network loadings were extracted based on a survey of various consumers (such as industrial, commercial and residential loads). For buses without any recorded data, their load PMs can be estimated based on buses having the same nature of consumers. These data enable the analysis and modelling of network operating conditions, and ultimately improve the accuracy of distribution system optimization [14]. In [15], energy prices, DG outputs and load demands were obtained by the interaction between forecasts and predictive models. Furthermore, the renewable energy generation profiles, including PV and wind turbines, can be obtained from distribution system operator or customers. These profiles can be statistically analyzed to generate the PMs that can be used for constructing realistic network operation conditions for running distribution network optimization.

### 2.2. Uncertainty of Measurements

There are measurement errors/biases in any real measurements and PMs of various variables, such as voltage, active and reactive power and line impedance. The uncertainty of measurements can greatly affect the optimization performance in applications, such as SE and load flow. The impact of measurement uncertainties (including PMs) on network analysis has attracted great attention [16–18].

In [18], the study presented the fact that different measurement accuracies have a great impact on estimation accuracy. In [19], the study analyzed the influence of measurement errors on SE performance, and it presented how the estimated deviations of bus voltages can be improved when the load measurement accuracy is increased. In [20], the minimum measurements required to assure full observability was studied, and it is pointed out how the measurement uncertainty affects the SE performance. The study provides a straightforward suggestion on the maximum accepted uncertainty of measurements that is able to keep the estimation errors within thresholds. In [21], the impact of PMs obtained from new power profiles (such as PV and Combined Heat and Power systems) on the total SE error is analyzed. It also points out that uncertainties should be addressed in network analysis when using the estimated PMs. The accuracy of network analysis can be also greatly affected by the uncertainty of the network parameter. In [22], the paper provides an analysis on the influence of the uncertainties of network parameters on SE errors.

The uncertainties of different types of measurements can be obtained in different ways [23]:

(1) Real measurements: Usually the uncertainty of this type of measurements is determined by the tolerance of measurement devices. Usually the requirement of measurement accuracy complies with certain standards, and was specified already during the meter development/design stage based on the purpose of the applications. Thus, the measurement tolerance can be obtained by specific standards [24]. For instance, [25,26] specify the ranges of different classes of voltage transformers and current transformers, as well as their phase displacement. IEC61000-4-30 specifies the maximum allowed uncertainty of voltage and current measurements for classes A and B performance [27]. In [28], the range of voltage measurement tolerance was set based on both measurement and transformer uncertainty, and the tolerance of power measurement was set by considering both Current Transformers (CTs) and Voltage Transformers (VTs) tolerance [28].

(2) Pseudo-measurement: As for PMs, its accuracy mainly depends on the performance of the estimator/forecaster and the credibility of the information used for estimation. PMs have a larger uncertainty/tolerance than real measurements. Therefore they are usually given lower influence on decision-making. In [29,30], 20% to 50% of errors were considered in PMs. In [18,31], the authors specify the PM errors for load demand under different scenarios. Generally, the PM of real power is smaller than that of reactive power, as more data sources (such as energy bill and scheduling) are available for the estimation of real power. The estimation (or indirect measurements) of network parameters also have certain levels of uncertainty. In [2], a range of tolerance for line impedances is specified. In [32,33], the authors provide the uncertainty of short-circuit impedances of general transformers and On-Load Tap-Changer Transformer (OLTCT) respectively. Table 1 summarizes the tolerance of a list of critical variables used in power system simulation [34].

**Table 1.** Tolerance ranges for measurements and network variables [34]. PMs = pseudo-measurements.

| Index | Variable | Tolerance Range |
|:---:|:---:|:---:|
| 1 | Voltage measurement | [0.14%, 3.04%] with 3-sigma |
| 2 | Power measurements | [0.17%, 6.16%] with 3-sigma |
| 3 | PMs of active power | [10%, 40%] with 3-sigma |
| 4 | PMs of reactive power | [20%, 50%] with 3-sigma |
| 5 | Line impedance | [0, 20%] with 3-sigma |

## 3. Optimization-Based Distribution Planning

Network planning can be divided into three categories based on time scale: short term planning dealing with contingencies, medium term planning for network maintenance and long term planning

for network expansion and up-gradation. To achieve an optimal planning strategy, it can be formatted as an optimization problem. The optimization problem generally is defined in the following form:

$$\min_u f(u,x),$$
$$\text{s.t. } g(u,x) = 0; h(u,x) \leq 0 \text{ (including } u_{\min} \leq u \leq u_{\max}). \tag{1}$$

where objective function $f$ is related to either technical or economical aspects, $u$ is the decision variables, $x$ is the state variables and $g$ is a set of equations in the context of power systems.

### 3.1. Optimal Meter Placement

Direct meter measurement is critical because of its accuracy and high influence in network analysis results. In the literature, it has demonstrated that different meter placement could greatly impact on the network analysis performance and eventually the network operation [35]. Thus, it is important to allocate the meters optimally around the distribution networks, and the analysis of meter placement strategy is attracting more attention than ever. In the literature, studies mainly focus on improving network observability and minimizing estimation errors in order to improve the estimation across the network [29]. For instance, [36] studies the problem of placing additional physical meters to improve state estimation (SE) accuracy. This problem considers both the existing metering structure and quantified performance improvement by adding an additional physical meter. In [37], meter placement was studied to minimize multi-objectives, including the network configuration cost and estimation errors in SE. In [38], meter placement was studied to minimize the peak relative errors in voltage and angle estimation against specified thresholds. In [2], meter placement was studied to specify the minimum cost and data accuracy that are needed for SE.

To obtain optimal meter placement, various approaches have been used. In [2], a dynamic programming-based approach was used to choose the optimal placement of measurement devices for SE procedures. For state-of-the-art SE models, heuristic or suboptimal algorithms are used widely for optimal placement of measurement meters especially for the purpose of SE. In general in the design of an optimization framework, three aspects should be considered in order to efficiently and accurately search for the optimal (minimum or maximum) value [35]: (1) trade-off between exploitation and exploration; (2) proper integration of gradient information; (3) proper use of prior knowledge for constructing a solution space and guiding optimization searching. Though they have been investigated much in the area of optimization development, they are not sufficiently addressed in power system-related optimization applications. Especially for the third point, the network configuration can be integrated in a search environment in order to narrow down the search space and eventually improve the optimization accuracy and efficiency. The rest of this subsection mainly focuses on the current studies in terms of these aspects, and also provides guidance on the design and preparation of optimization search space in distribution optimization applications.

Topological observability, which is used to evaluate the sufficiency of given measurements in carrying out static SE, was studied extensively with graph theory [39]. For instance, spanning trees have been used for topological observability analysis of power systems [40]. It is believed that network configuration/topology as prior knowledge can be very useful for providing hints/information for optimization searching. In [41], the problem of maximizing topological observability was formulated as a combinatorial optimal meter placement problem, and a hybrid approach (i.e., the combination of ordinal optimization and tabu search) was used to reduce the solution space for searching. In this way, the efficiency of searching in optimization is significantly improved. In [36], an algebraic form of circuit representation model was proposed to represent SE errors, which presents a two-node system and its circuit representation. Based on the circuit representation, the problem can be considered as a mixed integer linear programming problem and can be tackled by linear programming algorithms. This approach integrates circuit representation in the process of searching and optimization. In this way the circuit configuration is utilized in solution/search space. These studies have demonstrated the benefits of circuit representation in enhancing SE performance.

In [35], the development of an optimization framework which is particularly tailored for optimal meter placement under the context of distribution systems was investigated. A cost-effective monitoring scheme using limited devices/meters was obtained by integrating network configuration in optimization searching. The network configuration was represented by spanning/search trees that were constructed on the basis of network configuration. The optimization starts searching from the root of the spanning three, and the next search route at the junction is selected based on the performance improvement (i.e., gradient) along the branches while integrating the uncertainly/probability of choosing other routes that do not have instant improvement. In this way, exploration and exploitation can be balanced properly. This approach can be also used or extended for other optimal meter/device placements. Particle swarm optimisation (PSO) is used in [4,35] to search along the trees because of its easy implementation, fewer parameters and fast converge, and it is particularly useful in applications where there is a huge search space and certain prior knowledge.

In [38], ordinal optimization was applied to seek the optimal set of meter placement to minimize estimation errors. Ordinal optimization is a useful approach to reduce the size of the search space. Prior to optimization searching, the search space is reduced by selecting potential alternatives from favored good designs. The approach [38] makes sure the potential solution space contains top 0.1% good options with 0.99 probability, which ensures a balanced trade-off between exploration and exploitation. This greatly decreases computation load while ensuring the performance of the final option by making the most of the prior knowledge.

## 3.2. Distribution Generations (DGs) Planning

With the preference of using renewable energy nowadays in electricity generation, a large number of DGs have been and will be integrated into distribution networks. DGs can support system operation with the delivered reactive power. There are a number of various DG technologies, such as PV, wind turbines and fuel cells. The DGs can be classified based on the types of power delivered (real and/or reactive power) and power factor (unity, leading or lagging) [42]. For instance, some DGs deliver real power at the unity power factor (PF), such as with PV or biogas; some deliver both real and reactive power at 0.8–0.99 leading PF, such as wind generators, tidal, wave and geo-thermal generators; some deliver only reactive power at zero PF, such as with a synchronous condenser, inductor bank and capacitor banks; some deliver reactive power but absorb real power at 0.80–0.99 lagging PF, such as Doubly Fed Induction Generators (DFIG) wind generation.

Proper DG installation (with the optimal size, location, number and types) can bring multiple benefits to the grids, including reduction on energy loss [43], power factor correction, increasing feeder capacity, improving the voltage profile and meeting the increased load demand. Vice versa, inappropriate DG installation may result in constraint violation and network instability. The selection of DG size and location is considered as a combinatorial optimization problem that can be formulated as Equation (1). The objective function can be designed to indicate the solution quality based on the concerns. For instance, appropriate voltage profile is one of the critical operation concerns in distribution systems. In this case, the objective function can be designed in a way to indicate the severity or financial assessment of the voltage related issues, such as with the voltage profile, power loss, line-loss reduction and environmental impact. This concern can be directly included in the objective functions, or sometimes can be transferred to economic presentation before being included in the objective function. In simple cases, the objective function consists of power loss and other related costs (e.g., investment costs) [1], as given below:

$$Min\left[\left(\sum_{i=1}^{NC} IC + CC \times Q_i\right) + PC \times \sum_{i=1}^{NB-1} PL\right], \tag{2}$$

where *IC* denotes the investment cost; *Q*, the compensated reactive power; *CC*, the related cost; *PL*, power loss and *PC*, the related cost per kWh. *NB* and *NC* are the numbers of network buses and compensators, respectively. Apart from Equation (2), various objectives have been studied and

used to formalize the optimization function, such as the reduction of power loss, reduction on power congestion, improvement of voltage profiles [44], enhancement of stability [45,46], reliability, loadability and flexibility of operation. In [47], a multiple objective model was proposed to minimize the loss and maximize the DG capacity simultaneously. In this case, the objective function consisted of monetary cost (e.g., cost of investment, DG operation and power losses) and technical risks (e.g., violation of loading and voltage constraints). Since various technical issues are involved in the presence of DGs in distribution networks, [48] proposed a multi-objective function which assesses the technical impacts of DGs on network reliability and power quality based on a steady state analysis.

To find the optimal strategy for the pre-defined objective, various conventional and artificial intelligence based optimizers have been applied successfully to generate DG planning strategies, such as linear and non-linear programming (LP and NLP), ordinal optimization (OO) [47], heuristic techniques [49,50], genetic algorithm (GA) [51], evolutionary algorithm [1] and hybrid techniques [52]. In [42], the optimization algorithms used for DG planning were classified into four categories, as provided in Table 2. Among these four categories, artificial intelligence has been used most, especially the genetic algorithms. Then it is followed by the conventional techniques and optimization techniques. Sometimes, combining different techniques can provide better results, such as hybrid algorithms which combine optimization techniques and artificial intelligence. In the literature, the conventional techniques are mainly applied for single DG type planning, and usually are not for multiple DG type planning [42]. Compared with conventional techniques, artificial intelligence (AI) techniques and their hybrid approaches are more suitable for optimal DG planning considering multiple perspectives, i.e., multi-objective optimization.

**Table 2.** Categories of various optimization algorithms [42].

| Rank | Categories | Algorithms |
|---|---|---|
| 1 | AI techniques | genetic algorithm (GA), particle swarm optimization (PSO), tabu search (TS), fuzzy logic (FL), ant colony search (ACS), artificial bee colony (ABC), artificial neural network (ANN), simulated annealing (SA) |
| 2 | Conventional technique | residues, modal index techniques, eigen values, eigen vector, |
| 3 | Optimization techniques | dynamic programming, linear programming (LP), non-linear programming (NLP), interior point method, ordinal optimization (OO), gradient search method |
| 4 | Hybrid AI techniques | GA + FL, GA + optimal power flow (OPF), GA + PSO |

*3.3. Power Quality Mitigation*

Increased penetration of non-conventional DGs and connection of more power electronics in existing distribution networks raise great challenges in providing sufficient quality of supply. PQ significance is already acknowledged by regulatory bodies and Distribution System Operator (DSO), and the awareness has risen in recent years among various stakeholders. Power quality (PQ) phenomena are considered as the reliability of the system from the utilities' perceptive, and they may interrupt equipment and manufacturing processes and result in great financial losses to end users and grid operators [53–56].

Though PQ consists of a wide range of phenomena, usually the studies in the literature focus mainly on a number of important PQ phenomena, such as voltage sags, unbalance and harmonic. Voltage sags attracted a great deal of attention in PQ studies, with their substantial financial loss caused by the frequent interruption to equipment and manufacturing processes [57,58]. Voltage unbalance also has become more important than ever because of the continuously increased installation of one-phase- or two-phase-connected DGs or storage [59]. Unbalance phenomena cause thermal stress to equipment, and result in additional power loss and reduced efficiency of network operation [30,56]. With increased power electronic interfaced generations and non-linear loads in the systems, the harmonics phenomenon

is gaining increased attention because of increased thermal stress, telephone interference, equipment mal-operation and damage under resonance phenomena [60].

In PQ optimization/mitigation, it is important to understand PQ requirement. A number of standards have defined the PQ requirement and evaluation techniques, such as requirement of ride-through capability in terms of voltage sags [61,62], voltage characteristic recommendation [63], measurement accuracy requirement [27] and the required harmonics performance in distribution grids [64]. Because of the heavy penalties on violating standard specification, compliance with the standards at all times is required to avoid financial losses to both grid operators and end-users.

PQ mitigation can be defined as an optimization problem and solved by following the general procedures of the optimization framework given in Figure 1. With the identified PQ phenomena, the mitigation schemes for the corresponding PQ phenomena should be selected first. Various mitigation schemes have been explored in the literature to ensure the provision of appropriate PQ levels [65,66]. PQ phenomena can be mitigated from the equipment level to the network level. PQ mitigation of real time compensation can be implemented with power devices and harmonics filters, thanks to the advanced technology in power electronics, especially Flexible Alternative Current Transmission Systems (FACTS) devices which are able to adjust voltage, current and impedance to a certain extent. FACTS have undisputed mitigation capabilities and promising benefits in the long term [67–72], and have already been widely investigated for power system applications [73–75]. Alternatively, PQ phenomena can be mitigated through a higher level using prevention rather than cure. Rather than installing costly power electronic-based devices, network-based mitigation uses existing network resources in an effective way to resolve PQ issues, such as tree trimming schedules, for example. Network-based mitigation presents its benefits in network level PQ mitigation. After the mitigation schemes are selected, they will be made available in a solution pool for selection in the optimization process.

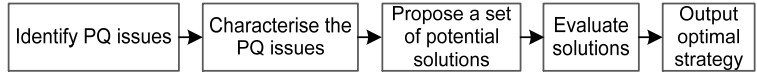

**Figure 1.** Illustration of power quality (PQ) optimization/mitigation [76].

To assess the appropriateness of the mitigation schemes/solutions, the objective function of optimization can be defined in a way to assess the mitigation effect of each potential solution against standard and compatibility levels. The objective function can also include the financial aspects of the mitigation solution. PQ performance can be assessed by proper indices, such as the bus performance index (BPI) which has been used to assess the practical impact of voltage sags on system operation [77], total harmonic distortion (THD) for harmonics phenomena and voltage unbalance factor (VUF) for unbalance [76]. With the PQ indices and their corresponding requirements, the objective function can assess the distance between poor PQ performance and its thresholds. If the PQ phenomena are considered individually, [78] proposes a number of PQ indices to define the gaps in terms of the three aforementioned PQ phenomena respectively, as given in Equations (3) and (4):

$$SGI = \sum_{j=1}^{B} \left| BPI_j - BPI_{\text{TH}} \right|_{BPI_j > BPI_{\text{TH}}}, \tag{3}$$

$$HGI = \sum_{j=1}^{B} \left| THD_j - THD_{\text{TH}} \right|_{THD_j > THD_{\text{TH}}}, \tag{4}$$

$$UGI = \sum_{j=1}^{B} \left| VUF_j - VUF_{\text{TH}} \right|_{VUF_j > VUF_{\text{TH}}}, \tag{5}$$

where $j$ is the bus index and $BPI_{\text{TH}}$ denotes the threshold.

The application of power electronic-based devices usually focuses on mitigating one particular PQ issue [67–70]. However, the installed devices usually impact more than one PQ phenomenon. Thus it is important to consider multiple related but critical PQ phenomena at the same time in order to improve efficiency. This will greatly reduce the investment cost in comparison to the case of

tackling each PQ phenomenon individually. In this case, the objective function can be designed in a way that it presents the comprehensive assessment of the impact of a solution on multiple critical PQ phenomena. Different ways have been adopted to aggregate the performance of critical PQ phenomena [79,80]. The aggregation methodology of the analytic hierarchy process (AHP) is useful if the optimization purpose is to keep the received PQ performance within certain requirements/standards. The AHP-aggregated PQ performance can be defined as Equation (6) below if the aforementioned three PQ phenomena are considered simultaneously:

$$UBPI_j = \text{AHP}\left(BPI_j,\ THD_j,\ VUF_j\right). \tag{6}$$

Given *UBPI* and the expected aggregated UBPI (i.e., $UBPI_{\text{TH}}$), the objective function in optimization can be defined as [78]:

$$PQGI_{\text{UBPI}} = \sum_{j=1}^{B} \left|UBPI_j - UBPI_{\text{TH}}\right|_{UBPI_j > UBPI_{\text{TH}}}. \tag{7}$$

The objective function given above integrates the PQ performance first before comparing it with the aggregated thresholds. Alternatively, the objective can be defined in a more specific way by considering the individual PQ performance against their limitations, as given in Equation (8), where each PQ phenomenon is compared with its corresponding threshold before the integration process:

$$PQGI_{\text{IND}} = \sum_{j=1}^{B} \text{AHP}\left(\left|BPI_j - BPI_{\text{TH}}\right|_{BPI_j > BPI_{\text{TH}}}, \left|THD_j - THD_{\text{TH}}\right|_{THD_j > THD_{\text{TH}}}, \left|VUF_j - VUF_{\text{TH}}\right|_{VUF_{i,j} > VUF_{\text{TH}}}\right). \tag{8}$$

With the defined objective functions, various optimization approaches have been investigated to search for the optimal PQ mitigation strategy, including the methods listed in Table 2. The greedy algorithm was adopted in [81,82] to solve large-scale optimization problems in distribution networks because of its benefit of simple implementation and relatively lower computation load.

*3.4. Addressing Uncertainty in Optimization Process*

The studies on the impact of network uncertainty on network analysis and optimization have demonstrated that the improved understanding of the uncertainty of network operating conditions is beneficial to the assessment of network performance [34]. Therefore, uncertainties should be considered when assessing the quality of the solutions in optimization in order to generate a better planning strategy. Apart from the measurement uncertainties mentioned in Section 2.2, uncertainty also exists in operating conditions, network parameters and topologies.

(1) Uncertainties in operating conditions: The operation scenario varies throughout the whole year because of factors such as different DG outputs and the loading of different types of customers. This can be addressed by using historical data to generate the simulation conditions that approach the actual operating conditions. The electricity consumption patterns of different types of loads can be obtained from a survey [13]. The generation profiles of renewable energy, such as PV and wind turbines, can be estimated based on weather, or obtained from realistic output [83]. In [84], actual varying loading, PV and wind profiles in different counties in Europe for the past decade are provided. It provides a wide range of data for power system modelling and uncertainty analysis. Regarding PQ simulation, there is uncertainty of factors such as the fault rate and harmonic injection. These should be also considered when assessing the PQ performance [78].

(2) Uncertainties in network topologies: Network topology should be provided in certain network analysis (such as with load flow and SE). The uncertainties of the frequent topology changes exit in distribution systems because of the operation of switching, which is adopted by grid operators to optimize the electricity provision even with the occurrence of outages. Without proper network topology, the analysis results are not accountable. The uncertainties of network behaviors and topologies on SE accuracy have been analyzed in [18,85], respectively. In [86], the uncertainty of

network configuration was reduced using a recursive Bayesian approach together with utilizing the SE outputs.

(3)　Uncertainties in network parameters: Network parameters are usually not given directly, and their values can be estimated via indirect measurements or estimation. Thus, uncertainties exit in these estimated network parameters, such as line impedances [2], short-circuit impedances for transformers [32] and OLTCT impedance [33]. In [87], a method based on the artificial neural network (ANN) and topology observability is used to evaluate the parameters which are missing in power systems.

The aforementioned uncertainties, including Section 2.2, have a great impact on the final planning/operation strategies obtained from optimization. To construct realistic operating conditions for network analysis, the uncertainties/tolerances of measurements and network parameters have to be taken into account in distribution optimization. Depending on the problems to be solved, the types of uncertainties that should be considered vary in different cases. In DG planning, the uncertainty of network conditions, especially the uncertainty of DG outputs, should be considered during the assessment of the appropriateness of the planed strategies. The uncertainty can be addressed in simulation-related settings and conditions, such as loading, power injection and consumption. The stochastic programming model, which constructs the probability distribution of uncertainties based on historical data, is widely used to model the uncertain operating conditions in power system simulations. The data-driven modeling approach is used in [88] to model uncertainties for reactive power optimization in active distribution networks. In [89], the uncertain demand is characterized by probability and possibility distribution using stochastic optimization methods. In probability-based approaches, the uncertainty related studies mainly consider expected values and standard deviation. The tolerances in Table 1 can be transferred to standard deviation. For example, if the deviation or tolerance from the mean $\mu$ is given, the standard deviation of the distribution of the measurement can be calculated by $\sigma = \frac{\mu \times \%\text{error}}{3 \times 100}$ [29]. Apart from the approaches mentioned above, fuzzy approaches are adopted to address the uncertain load modeling, voltage constraints and thermal constraints [47]. Depending on the optimization purpose, sometimes only the worst case scenario (robust programming) of generation dynamics is considered. Monte Carlo approaches take into account all possible scenarios of the network operating conditions [2]. However, with the large number of possible operation scenarios, it is costly in computation if all of them are simulated in Monte Carlo approaches, as, in general, many iterations are needed to yield the final optimal solution in the optimization process. Thus, representative operation conditions can be probabilistically sampled/selected in order to yield the most likely assessment. Clustering-based approaches can be used to select the representative operating points, considering the fact that operating conditions repeat seasonally or yearly in actual cases. For instance, the same types of loads repeat certain patterns to some extent, and DGs also have similar varying trends. Thus, the operation points can be sampled to obtain a number of representative operating points that can roughly cover essential operating points throughout the whole year. In [78,90], clustering was used to select operating scenarios for studies. Inputs to the clustering consist of the load and DG profiles of different types. There are various clustering approaches that can be selected, such as fuzzy c-means, K-means and the agglomerative clustering algorithm. After obtaining the clusters, the centers of the clusters can be taken as the representative operating points and used for simulation in optimization.

## 4. Optimization-Based Distribution Operation and Management

### 4.1. Constraint Management

The active distribution networks nowadays are exposed to frequent constraint violation, mainly due to the continuously increased loading and the more intermittent nature of DGs. Constraint violation is costly to both utilities and end-users in the form of penalty or purchasing new equipment. Utilities have the responsibility to keep the grid conditions within the limits at all times. Therefore,

constraint management is attracting great attention because of the network changes as discussed in Section 1.

Constraint management involves various aspects in network operation, such as voltage constraint and fluctuation. Because of the increased loading and preference of deferring system expansion, congestion issues are becoming inevitable in peak time [91]. The constraints can be given by strict standards. For instance, the voltage variation range can be given by limits as −6.0%/+10.0% in distribution networks [92]. These constraints may vary in different countries and are usually set and enforced by relevant regulatory agencies. In some cases, relevant laws and regulatory acts are implemented for mandatory by governmental legislative bodies. Apart from the regulation sides, customers may require a different quality of services because of the diversity of customer natures, and some customers could expect stricter constraints than the service normally supplied [93].

Constraint management is designed to ensure the economic, efficient and coordinated delivery of electricity while meeting the requirements set by regulations or customers. The constraints can be managed with the help of optimization, and the objective functions can be constructed to suggest the severity of constraint issues or its transferred economic cost, while the variables could be set as the sources that can be utilized to solve the constraint issues. The recent trend in constraint management research is the management of the critical loading condition rather than in increasing network capacity, which is costly. The management of the critical loading condition can be implemented by optimally utilizing the available resources in the network, especially with the advanced technology in the development of smart grids (e.g., the widespread installation of distributed energy resources, the active engagement of customers, the availability of increased flexibility exchange offered by customers and the recent advances in Information and Communication Technology (ICT). Utilities should make sure that the system operates within given limits, while making the most of the services provided by potential providers or tenders [94,95].

Different types of grid-related optimization are documented in the literature for constraint management. One of the categories is based on optimal power flow (OPF). In [96], OPF was used to minimize a multi-objective function which was to reduce the power losses and minimize the risk of overloading and voltage violation. Constraints of both decision variables and network states were implemented by inequality constraints, and the unbalanced power-flow equations were enforced by equality constraints. This management of thermal and voltage constraints can be implemented by properly dispatching reactive power of DGs, curtailing generation or coordinating tap changers [96,97], which can be set as stochastic decision variables in optimization.

Various optimization algorithms were adopted for OPF problems. Classical optimization algorithms such as linear programming (LP) are widely used for solving OPF problems [98]. In [99], the capacity constraint was used to form linear inequality constraint for the objective function in OPF. Heuristic methods have been investigated for these problems, such as particle swarm optimization [100], ANN and GA [101]. One of the main concerns of heuristic methods is the uncertainty of the optimality and heavy computation load if a large number of iterations are required for convergence. Usually different techniques were integrated in the heuristic methods in order to jump out of the local optimal solutions and obtain the global optimal solution. For instance, group search optimization is applied to generate a better compromised solution in problems with a multi-objective nature [96].

Alternatively, SE can be used for constraint management [102]. One of the most challenging factors in applying SE in constraint management is formulating the optimization problem while addressing the operating limits in control variables. The SE problem is defined as:

$$\min_{X}\Big[[Y - H(X)]^{T} R^{-1} [Y - H(X)]\Big] \tag{9}$$

where $X$ and $Y$ are state variables and the set of measurements, respectively, $R$ is the covariance matrix of measurement errors and $H$ denotes the nonlinear power system equations.

The SE in transmission networks can be conducted using single phase analysis, as transmission networks are considered being balanced. However, this assumption is not valid in distribution networks, and three-phase SE is needed, as unbalance phenomenon is one of the most common PQ phenomena in distribution networks. SE in a distribution network can be solved by distribution system SE (DSSE). The missing data at non-monitored busbars can be compensated using PMs, which enable the state of an unobservable system to be estimated. DSSE is very similar to the process of performing probabilistic load flow. Figure 2 presents the SE processes, in which the un-observed branches can be identified using an observability analysis tool. This, too, will suggest the required but missing information, which often can be obtained from PMs.

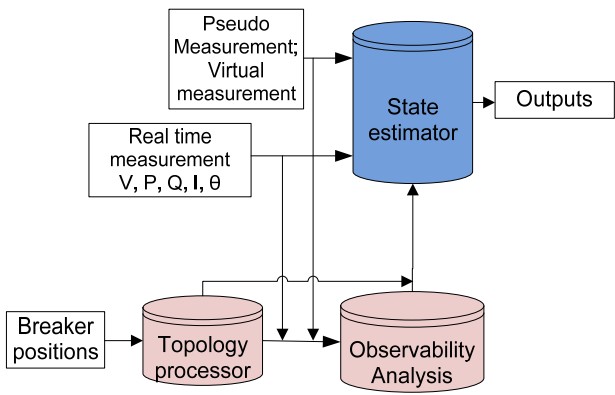

**Figure 2.** Illustration of state estimation (SE) processes.

Load flow studies mainly focus on establishing long term variation in network parameters, whilst DSSE aims at establishing the current system state. Both techniques use Newton's method and aim at estimating the statistical variation of parameters based on uncertainties. Notable works on this topic include [103], which defines the structure for three-phase load flow, and [104] which estimates the variation of network parameters with the existence of uncertain wind generation.

Different uncertainties mentioned in Section 2.2 have different influences on the final optimal solutions. In [34], an analysis of the impact of different uncertainties on SE locally and globally is provided, and the results show that the estimation performance varies significantly when addressing the uncertainty in a different way, and, furthermore, the performance varies when variables are with a different uncertainty range. Another important aspect of three-phase DSSE and three-phase probabilistic load flow studies is the correlation between measurement errors [19]. Correlation of measurement errors could be incorporated into a three-phase DSSE formulation using a generalized least squares (GLS) approach [105] in order to minimize the adverse impact caused by uncertainties. Correlation in multi-phase networks is covered in more detail in [103,104,106]. The weights of different uncertainties in the optimization framework can be properly assessed and set, such as with *R* in Equation (9). Furthermore, in network operation, it is important to continuously update the pseudo-measurements and their predictive models in order to minimize the impact of uncertainties on decision-making. In [15], the prediction models of various variables are updated constantly via self-correction, which reduces the prediction errors. This approach can provide more accurate predictions for the model predictive controller (MPC) to generate control actions.

### 4.2. Demand Side Management and Flexibility Exchange

With the increased flexibility for control in active distribution network and the fast development of communication technologies in smart grids, flexibility exchange between unities and demand-sides is becoming feasible and getting more attention. Demand-side management (DSM), DGs and storage are taken as essential elements for smart grid development, and, more promisingly, can facilitate grid operation/management [5]. DSM can be used to participate in constraint management,

which is discussed in Section 3.1. DSM was studied for different applications, such as shifting load [107] and congestion issues [108–111]. In [112], a decentralized approach was proposed to control DG power outputs in order to implement real-time management of thermal constraints and voltage issues. A distributed cooperative optimization operation strategy was proposed in [113] to achieve the cooperative operation of DG and flexible loads in active distribution networks. In [114], the flexibility exchange strategy was developed to tackle congestion issues and maintain acceptable voltage profiles, while having the minimum contribution from customers or aggregators. In this approach, two optimization processes were applied. One was to minimize the difference between the network state and the expected states. Voltage profiles and power flow were tuned towards the expected state by optimization and network estimation. The optimization objective function can be constructed based on general SE error as defined in Equation (1), while $Y$ is set to the expected network states and $R$ is replaced with the coefficient of variable flexibility, which indicates how much the network state can deviate from the expected values. To achieve this aforementioned purpose, the optimization objective is defined as [114]:

$$
\begin{aligned}
F_{\text{optimisation}}(R) = \sum_{i=1}^{N} \Bigg( & \sum_{j=1}^{K}\left|\left(P_{ij,\text{adj}}(R)-P_{ij,\text{ori}}\right)\right| + \left|\sum_{j=1}^{K}\left(Q_{ij,\text{adj}}(R)-Q_{ij,\text{ori}}\right)\right| \\
+\beta \times & \left(\sum_{j=1}^{K}\left|P_{ij,\text{adj}}(R)-P_{ij,lim}\right|_{P_{ij,\text{adj}}(R)>P_{ij,lim}} + \sum_{j=1}^{K}\left|Q_{ij,\text{adj}}(R)-Q_{ij,lim}\right|_{Q_{ij,\text{adj}}(R)>Q_{ij,lim}} \right) \Bigg)
\end{aligned}
\tag{10}
$$

where $\beta$ is a Lagrange multiplier and $N$ and $K$ represent the total number of buses and phases equipped with a flexibility exchange function. $P_{ij,\text{ori}}$ and $Q_{ij,\text{ori}}$ are real and reactive power (P and Q) before flexibility exchange and $P_{ij,\text{adj}}$ and $Q_{ij,\text{adj}}$ are the P and Q after exchanging flexibility. This problem can be solved by the Newton-Raphson approach. Genetic algorithm-based optimization approaches in Table 1 can be also selected for this type of optimization problem.

The optimal flexibility exchange can be implemented by a large scale of distributed customers/stakeholders via smart pricing market or exchange platform. In this way, incentives or penalties in the flexibility supply chain can be used in real time to influence customers' electricity use [115]. On the other hand, the flexibility exchange strategy can be also used for the determination of electricity prices. DSM functionality integrated with pricing strategy has also been studied for other applications, such as maintaining good operating conditions [111,112,116].

When planning DGs (introduced in Section 3.2) for the operation of DSM and flexibly exchange, both distribution planning and operation should be considered. Though in general distribution planning and operation are discussed separately in the literature, these two topics are closely related. The planning to some extent is to improve/facilitate the operation. Thus, the operation should be integrated in planning process (especially in optimization process) when assessing the performance of the planning strategy. In other words, the operation performance with and without the planning strategy should be assessed during the planning stage. Therefore, the optimization process in general should take both network planning and operation into account so that the optimal planning strategy can be comprehensively assessed.

## 5. Future Distribution Networks

Distribution networks in future will be more complex, flexible, controllable and smarter [23]. It will be more active in operation with the large scale of DGs and different formats of demand response schemas from customers' engagement. In terms of technologies, there will be a variety of generation techniques, power electronics devices and a large number of dynamically controlled elements. With highly active engagement from a variety of stakeholders, the electricity market will play an important role in influencing behaviors of various stakeholders in order to achieve the goal of reducing running cost and providing efficient and stable services.

*5.1. Big Data and Challenge*

The distribution networks will highly rely on monitoring, and more data/information with different frequencies will be available, including smart meter readings, billing information, the profiles and preferences from different stakeholders and DSM schemes. They can be explored for different purpose, such as in operation, pricing, control, topology analysis and the improvement of DSSE and load estimation. The future high data streams suggests that communication infrastructure should be enhanced to facilitate these changes. To avoid the high cost of investing in communication infrastructure, the data can be sent only when it is triggered by certain events. For instance, data are collected and communicated for SE analysis when there is an indication of potential issues [14,102]. To avoid being drowned in a huge data pool, big data analyst approaches can be used to extract useful information such as correlation information. Optimization in utilization with big data analysis and artificial intelligence will be more extensively investigated in the future, and it can also reduce the amount of transferred data if the data are analyzed locally.

At the same time, the operation and management of future distribution networks will face great challenges because of the uncertain network operating conditions and uncertain data accuracy. For this issue, the development of a closed-loop information flow framework is appealing for the future, and it can be used to self-correct the inaccurate information. Figure 3 illustrates the use of close-loop information flow for DSSE. The DSSE results and estimation can be fed back to the load estimation, demand profile and network topology analysis in order to auto-correct the input data. The process continuously updates and improves the information prediction using estimation results. Thus, the operation conditions can be better predicted. Future network analysis can explore on artificial intelligence more for network operation, in order to achieve a smarter grid with a fast response to changes network in networks [14,23].

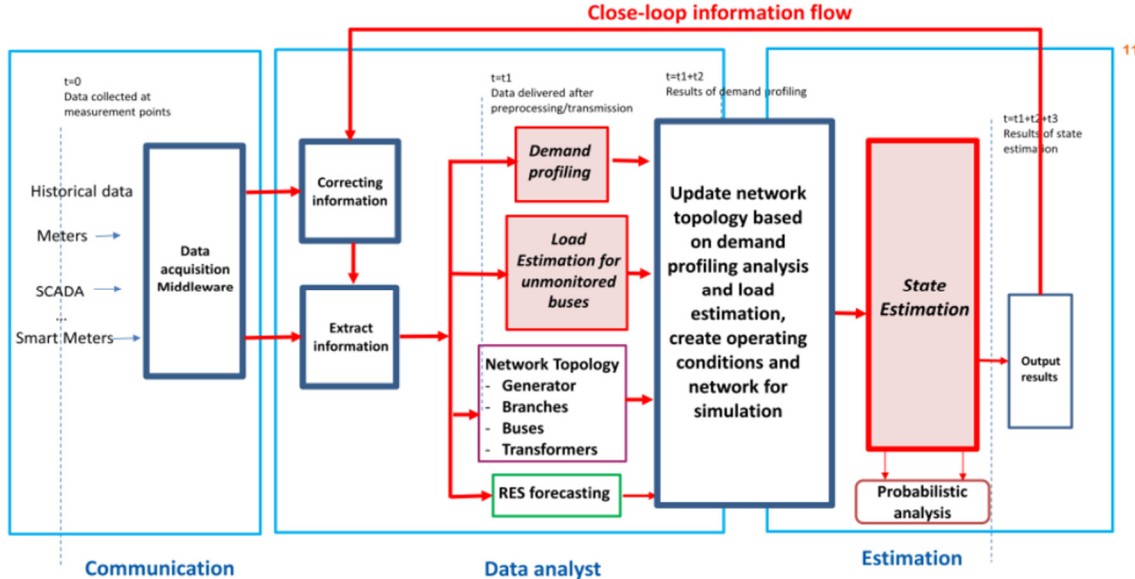

**Figure 3.** Illustration of data and information flow in network analysis.

*5.2. Integrated Distribution Optimization between DSO and TSO*

The interaction between DSO and TSO will be required to achieve a full integration of system planning and operation in the future. With more flexibility/resources in distribution networks that can be utilized for TSO operation, the flexibility exchange between both operators will definitely tend to be more in the future, and the future distribution optimization should consider more of the integration and interaction between TSO and DSO sides. Taking the DG planning as an example, the increased installation of DGs in distribution level can influence the transmission network operation.

DGs, with the characteristics of intermittency and disturbances, can potentially cause power imbalance issues, which is the responsibility of TSO. On the other hand, DGs with reactive power support can be used to provide flexibility exchange sources to TSO in order to maintain voltage profiles [117]. Given the potential impacts of DG units on operation in a transmission network, DSO should plan DG connection while considering the TSO side, such as infrastructure adequacy and capacity. In [117], a practical case in Portugal is provided, which suggests the necessity of interaction between DSO and TSO in DG planning to avoid potential adverse issues. It shows that extra information is extremely important for optimization-related applications in Distribution Management System (DMS) and Energy Management Systems (EMS) functions for both TSO and DSO.

### 5.3. Differentiated PQ Supply

Considering efficiency and cost-effectiveness, it is not necessary to provide excess PQ quality that is beyond the requirement from customers, and it not optimal from the perspective of utilities to improve the power quality for the entire grid. Thus the promising solution for future distribution network is to address the problem locally/zonally. With the increased flexibility and controllability of distribution future networks, it is possible to provide customers with different PQ levels. The PQ requirement can be determined by the effect of PQ disturbances on their activities, using the electro-economic nature of methodology [118], in which zonal PQ thresholds are determined by the nature of customers and the sensitivity of their equipment to PQ phenomena. The differentiated PQ provision can be implemented by premier contracts [119] or other similar strategies. In this case, the optimization takes into account the differentiated PQ requirements in different zones and generates the optimal PQ mitigation strategy that provides required supply of service in a cost-effective way.

Taking the three PQ phenomena mentioned in Section 3.2, the objective function in Equations (7) and (8) can be modified to the following functions to address the zonal PQ thresholds [78]:

$$PQGI_{\text{UBPI}} = \sum_{i=1}^{N}\left(\sum_{j=1}^{B_i}\left|UBPI_{i,j} - UBPI_{\text{TH},i}\right|_{UBPI_{i,j}>UBPI_{\text{TH},i}}\right), \tag{11}$$

$$PQGI_{\text{IND}} = \sum_{i=1}^{N}\left(\sum_{j=1}^{B_i}\text{AHP}\left(\left|BPI_{i,j} - BPI_{\text{TH},i}\right|_{BPI_{i,j}>BPI_{\text{TH},i}}, \left|THD_{i,j} - THD_{\text{TH},i}\right|_{THD_{i,j}>THD_{\text{TH},i}}, \left|VUF_{i,j} - VUF_{\text{TH},i}\right|_{VUF_{i,j}>VUF_{\text{TH},i}}\right)\right), \tag{12}$$

where $THD_{\text{TH},i}$ represents the THD thresholds in zone $i$.

### 6. Conclusions

Distribution networks are becoming more complicated and flexible with increased controllable elements/resources and active engagement from various stakeholders. This suggests that distribution planning and operation are becoming more complicated and need to take into account different aspects, such as the constraints of the new controllable variables and the customers' availability in participating in network operation. Meanwhile, there are also new opportunities for solving distribution operation issues. For instance, power electronics interfaced renewable energy generation can be used to assist network operation with their capability of controlling reactive power. The loads will become more controllable and can be integrated in network operation or constraint management via a proper flexibility exchange platform. To enable cost-effective operation and management in distribution networks, the strategy of utilizing the available resources/flexibilities should be carefully planned using a tailored optimization process, and at the same time account for various constraints and uncertainties of the network behaviors. This paper suggests what should be considered in the design of optimization frameworks. For instance, in distribution optimization frameworks, the network configuration as prior knowledge should be integrated in searching space in order to improve the optimization accuracy and efficiency. Furthermore, to generate appropriate optimal planning and operation strategies, it requires realistic operating scenarios for network performance assessment. To construct accurate operating conditions for optimization, it is important to have sufficient data even with the existence of uncertainties. With the huge amount of detailed data becoming available in a smart grid, data analytic

and artificial intelligence approaches can be used to extract useful information to construct more accurate network operation conditions for optimization. The differentiated uncertainties of various data sources should be considered and addressed in the optimization frameworks. The uncertainty can be also minimized by developing techniques to improve/correct uncertain/missing data using a close-loop data flow framework.

Though the distribution operation will be more complicated, the development of proper distribution optimization techniques will enable the network operation running in a more cost-effective way, with reduced running cost, improved operation efficiency, better constraint management and a satisfactory quality of services.

**Funding:** This research received no external funding.

**Conflicts of Interest:** The author declares no conflicts of interest.

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
