# Peer review of "Review on Distribution Network Optimization under Uncertainty"

_energies, doi:10.3390/en12173369_

Round 1
Reviewer 1 Report
This paper discussed an interesting topic. However, the paper needs to be improved based on the following comments:
1- Authors need to present in better way the benefits of having a review paper Distribution Network Optimization under Uncertainty
2- Rewrite the introduction section in better way and showing the main aim of this paper and the story behind it.
3- There is more techniques and method used than that mentioned in table 1 or the paper such as MPC (Model predictive control) or SMPC.
4- Where is the predictive controller with load forecasting?
5- latex error in pages 8 and 11. Please check carefully your paper
6- Renumbering the sections and subsection (1.1 , 1.2,…..)
7- Why only some of the techniques are dissuaded in paper not all of them
8- The authors need to show what the relation between sections I , II and III section 3.2.
9- It is significant to show the advantages and disadvantages of each technique.
10- The authors need to extend the discussion section and present previous work in better way by comparing the papers results and show how the advantages and disadvantages of each technique reflect on the results and compare between them.
Reviewer 2 Report
While I understood the overall goals of the paper, there are some major issues that need to be adjusted for publication, which are as follows:
- A clear & short description of the content of the sections that the paper will address forwards is missing in the introduction section, which is usually placed in the last paragraph of the introduction. I was expecting that I will be introduced to the section titles for better organization of the paper.
- A relation between all the addressed topics in the paper is missing (optimal energy meters placement, improving optimization efficiency and accuracy, future network optimization issues,…etc). Why are these topics are addressed together? Where does this paper stands in comparison with other review papers? What added value does the paper give to the reader?
- Throughout the paper there is a problem with all figure references in the text. For example, lines 146, 159, 238, etc...
- Equation 10 was very difficult to read (very small font size) and no descriptions of the variables and parameters of the equation is presented.
- The conclusion part is weak in its presentation and feels rushed. Rather than just writing again what is carried out in the paper, I was expecting clear "conclusions", take aways & reflections from the author’s point of view regarding the issues addressed in the paper.
- I suggest that the author include more recent references (There are only 2 references from 2018, 5 from 2017, none from 2019). I believe that more updated references must be included for the benefit of the paper.
- The paper needs an extensive editing of English language.
Round 2
Reviewer 1 Report
The authors addressed all comments and no further comments.
Reviewer 2 Report
No Comments